# A Review of Paper-Based Sensors for Gas, Ion, and Biological Detection

Phillip Nathaniel Immanuel [1], Song-Jeng Huang [1], Yudhistira Adityawardhana [1,*] and Yi-Kuang Yen [2,*]

[1] Department of Mechanical Engineering, National Taiwan University of Science and Technology, Taipei 10607, Taiwan; nathaniel.philip@gmail.com (P.N.I.); sgjghuang@mail.ntust.edu.tw (S.-J.H.)

[2] Department of Mechanical Engineering, National Taipei University of Technology, Taipei 106344, Taiwan

* Correspondence: yudhis1996@gmail.com (Y.A.); ykyen@ntut.edu.tw (Y.-K.Y.)

**Abstract:** Gas, ion, and biological sensors have been widely utilized to detect analytes of great significance to the environment, food, and health. Paper-based sensors, which can be constructed on a low-cost paper substrate through a simple and cost-effective fabrication process, have attracted much interests for development. Moreover, many materials can be employed in designing sensors, such as metal oxides and/or inorganic materials, carbon-based nanomaterials, conductive polymers, and composite materials. Most of these provide a large surface area and pitted structure, along with extraordinary electrical and thermal conductivities, which are capable of improving sensor performance regarding sensitivity and limit of detection. In this review, we surveyed recent advances in different types of paper-based gas, ion, and biological sensors, focusing on how these materials' physical and chemical properties influence the sensor's response. Challenges and future perspectives for paper-based sensors are also discussed below.

**Keywords:** review; sensors; paper-based; performance; properties





## 1. Introduction

Technological developments in the world affect human life in ways that cannot be easily separated from industrialization. Many industries are becoming part of human life, such as manufacturing, natural gas, petrochemicals, and food processing. Although industries can impact human life, there are problems caused by industries that can affect the ecosystem and health, such as toxic gases, inflammable chemicals, and pollutants. To ensure the safety of the living environment, it is crucial to continuously monitor the trace-level concentrations of gases and ions released by industries into the environment.

Toxic gases, such as nitrogen dioxide ($NO_2$), ammonia ($NH_3$), carbon monoxide (CO), and volatile organic compounds (VOCs), including ethanol, toluene, and acetaldehyde, can be harmful to humans and cause air pollution. Exposure to these toxic gases and compounds can lead to respiratory illnesses, damage to the immune system, and lung abnormalities [1,2]. Additionally, certain heavy metal ions typically found in water, such as mercury (Hg), lead (Pb), and cadmium (Cd), are dangerous to human health. These metals tend to form complexes with the ligands found in sulfur-, nitrogen-, and oxygen-containing biological and chemical substances. This can lead to changes in the molecular structure of proteins, inhibition of enzymes, and hydrogen bond breakage. These complex heavy metals are probable carcinogens and have adverse effects on the human body, including the central nervous system (CNS), kidneys, and liver [3,4]. To monitor these problems, a sensor is needed to detect them.

Sensors are often used in real-time applications as they are selective, sensitive, easy to make, small, work better at lower temperatures, and use less power. They are made from semiconductor materials and generally use carbon-based nanomaterials, conductive polymers, metal oxides, and inorganic chemical compounds. These materials have applications

in sensing gases and ions [2,4,5]. In addition, several biological samples and molecules, including human urine, blood serum, blood plasma, and cholesterol, can be detected by sensors [6].

There are several types of sensors, such as semiconductor-based sensors, polymer-based sensors, silicon-based sensors, and paper-based sensors [7–10]. Paper-based sensors offer several benefits over the other types, including their flexibility, affordability, lightweight nature, environmental friendliness, degradability, and renewability. Despite being commonly used for writing, printing, drawing, and packaging, paper has demonstrated its effectiveness as a material for sensors and devices in analytical and chemical applications. This is because paper acts as a dielectric material, providing excellent insulation properties [11,12].

The physical properties of paper-based sensors can be observed through the morphology of their material and/or paper and surface characteristics. The chemical properties can be determined by the interaction between the sensing material and the contact element and/or between paper and the sensing material. Either the physical or chemical properties, or both, can have an impact on the sensing performance. This performance can be measured by the sensitivity and limit of detection (LOD) of the sensor. Further, there are also some challenges in the future development of paper-based sensors. In this review, we report on paper-based sensors and how their physical and chemical properties influence their sensing performance.

## 2. Background

Generally, the effectiveness of paper-based sensors depend on several factors such as sensitivity, response time, recovery time, and the limit of detection. Sensitivity pertains to the sensor's ability to detect the specific chemical species within the desired range of interest. Response time is usually stated as the amount of time needed to reach 90% of the final value, measured from the time that the measured variable's step input changed. Recovery time is the time taken by the sensor to return to its initial value after the target gas concentration has decreased to zero. The limit of detection (LOD) is determined by a statistical model that utilizes both simulated and experimental data, and is calculated based on a calibration curve [13–15].

### 2.1. Performance Factors

The paper-based sensor's performance requires consideration of several factors. Gas sensor sensitivity can be evaluated by examining the sensor's resistance-to-air ratio. Additionally, when dealing with oxidizing gases, sensitivity (S) can be calculated using Equation (1).

$$S = \frac{R_g}{R_a} \tag{1}$$

Conversely, for reductive gases, the sensitivity is represented by Equation (2), which is the inverse ratio of Equation (1). By employing these equations, the sensitivity of paper-based sensors can be effectively determined and analyzed.

$$S = \frac{R_a}{R_g} \tag{2}$$

where, $R_g$ is the sensor's resistance in the ambient environment, and $R_a$ is the sensor's resistance in the air. Both $R_a$ and $R_g$ are measure in Ohm units. The analytical signal can be effectively determined by evaluating the ratio between the resistance measured in the air medium and the resistance observed when the target gas is present in the air. This calculation is represented by Equation (3) [13,16].

$$S = \frac{R_a - R_g}{R_a} \tag{3}$$

The sensitivity of the gas sensor is also defined as the ratio of the resistance change to the baseline resistance (Equation (4)). In addition, the sensitivity of the gas sensor can also be calculated using current, where $I_a$ (A) represents the initial current in the air, and I (A) represents the current under target gas exposure (Equation (5)) [11].

$$S\,(\%) = \frac{R_g - R_a}{R_a} \tag{4}$$

$$S\,(\%) = \frac{I - I_a}{I_a} \tag{5}$$

The sensitivity of ion and biological sensors can be determined by measuring the change in resistance, which is expressed as the normalized values of resistances. In this measurement, $\Delta R$ represents the difference between the initial resistance (R-initial) and the final resistance (R-final). This calculation is represented by Equation (6) [17,18].

$$S = \frac{\Delta R}{R} \tag{6}$$

After sensitivity, the sensor's LOD is the next essential factor that determines the sensor's performance. LOD can be calculated using the ratio 3-sigma as the concentration standard deviation and m as the slope of the calibration curve, as shown in Equation (7). This equation can be used for gas, ion, and biological detection [19–21].

$$LOD = \frac{3\sigma}{m} \tag{7}$$

*2.2. Properties*

Sensitivity and/or limit of detection are important factors that influence the physical and chemical properties of a sensor [20]. In addition to the morphology of the material and/or paper, which is one of the physical properties of the paper-based sensor, several other physical properties of paper, such as porosity and surface area, can influence its sensing ability. Porosity refers to the proportion of empty space compared to the total volume of a membrane, indicating the capacity of a gas or liquid to pass through the paper surface. It plays a crucial role in determining the capillary flow rate, although it should be distinguished from pore size, a separate parameter. Porosity is also commonly referred to as the "void fraction," representing the amount of empty space within a specific material volume. Furthermore, processes such as surface sizing and coating can lead to fiber compression and sealing, ultimately reducing the porosity. It is important to note that porosity directly influences the permeability of paper [22].

The extent of a paper's surface has a considerable influence on various aspects, such as the way it is printed and applied onto substrates, as well as its sensing properties, which rely on features such as sensitivity. Calculating the surface area ratio involves multiplying the internal surface area ($m^2$/g) by the basic weight. When pore size increases, the surface area decreases in a non-linear manner, while it increases linearly with thickness and non-linearly with porosity if all other factors remain constant [22]. After porosity and surface area, surface chemistry and mechanical properties of paper should be considered. The surface chemistry of the paper substrate can affect the interaction between the sensing material and the contact or analyte. In addition, paper substrates can be employed for specific interactions and improve sensor selectivity with surface modifications [23]. Both flexibility and strength are mechanical properties of the paper substrate that can influence the stability and durability (time) of the sensor during use [24].

**3. Paper as Sensor Substrate**

Paper serves many important functions in human life, not only for common tasks such as writing and drawing but also as a substrate for sensors due to its inherently flexible

nature [25]. Paper as a substrate has many excellent properties compared to other substrates, including good flexibility, low cost, eco-friendly nature, low weight, and expandability [26]. In comparison to other substrates like glass and silicon, which lack the flexibility of paper and polydimethylsiloxane (PDMS), paper offers a cost advantage over both. Additionally, while PDMS exhibits good flexibility, it is comparatively more expensive than paper [26,27]. Further comparisons between different substrates are presented in Table 1.

**Table 1.** Comparison of paper as substrate to other substrates [28].

| Property | Paper | Glass | Silicon | PDMS |
|---|---|---|---|---|
| Surface Profile | Medium | Very Low | Very Low | Very Low |
| Flexibility | Yes | No | No | Yes |
| Physical Structure | Fibrous | Solid | Solid | Solid, Gas-Permeable |
| Surface to Volume Ratio | High | Low | Low | Low |
| Fluid Flow | Capillary Action | Forced | Forced | Forced |
| Biodegradability | Yes | No | No | To Some Extent |
| High-throughput Fabrication | Yes | Yes | Yes | No |
| Cost | Low | Medium | High | Medium |

There are several types of paper used as sensor substrates, including cellulose paper, filter paper, and others. Cellulose paper is derived from materials such as wood, straw, reeds, and waste paper. Furthermore, this type of paper is both cost-effective and environmentally friendly. Its high porosity, good roughness, biocompatibility, biodegradability, and hydrophilic properties make it suitable for various sensor applications [29,30]. Additionally, filter paper has excellent wicking ability, which makes it a popular selection for fabricating paper-based sensors. Whatman brand filter paper no. 1 is mostly used because it is a standard-grade filter paper with medium retention and flow rate. Besides No. 1, there is No. 4, a filter paper with larger pores compared to the standard grade, which is often selected to mitigate the potential restriction of capillary pores and hinders liquid penetration caused by the solvent-induced swelling of cellulose fibers [31]. Apart from cellulose and filter paper, chromatography paper is also an option. Chromatography is popular due to the smooth surface it exhibits; both sides are uniform, and it has a medium flow rate and a 0.18 mm thickness, which enables compatibility with commercial printing machines [28]. Both filter paper and chromatography paper are derived from cellulose [32,33].

Various techniques are employed to immobilize responsive materials in cellulose substrates, including dip coating, drop casting, and vacuum filtration. Achieving strong physical interactions or chemical bonds between the responsive material and the cellulose substrate is crucial for successful immobilization. Substrate modification might be necessary to enhance the retention of the sensing material. In the dip coating method, cellulose substrates are immersed in a solution containing the desired recognition element for a specific duration, followed by solvent evaporation. Drop casting is another commonly utilized technique, where a small volume of solution (e.g., 1–10 μL) is deposited onto the cellulose substrate. However, it is important to note that the capillary-driven flow on the cellulose substrate during drop casting can lead to uneven distribution of the responsive material [34]. There are other types of paper, such as glossy and array paper. Glossy paper is a substrate comprised of cellulose fiber and an inorganic filler [35,36]. In addition, an array of paper strips has been developed for the detection of several metal ions [37].

Several variables can be controlled to ensure the production of high-quality paper substrates. Firstly, the selection of paper type, such as cellulose paper and filter paper, is crucial, as each paper type possesses unique properties, including porosity and surface chemistry. Choosing the appropriate paper type based on the desired sensor performance allows for control over these variables. Secondly, surface modification such as coating can be applied to the paper substrate to enhance porosity, surface area, and surface chemistry, ultimately improving the sensor's performance. Lastly, the fabrication techniques employed in sensor production, such as dip coating, also impact the properties of the paper substrate. By controlling factors such as solution concentration, immersion time, and drying condition

during fabrication, one can effectively regulate the porosity, surface area, and surface chemistry of the paper substrate [38–40].

## 4. Fabrication

Fabricating a paper-based sensor requires several essential characteristics, including cost-effectiveness, simplicity, and efficient processing. Various techniques involving chemical modification and physical deposition are employed, enabling their versatility in diverse applications. Figure 1 illustrates fabrication methods such as photolithography, inkjet printing, and screen printing. Material choice and modifications depends on the selected fabrication approach. Notably, a significant body of research is dedicated to achieving precise liquid confinement within specific regions on the paper in the context of paper-based microfluidics.

In 2007, a novel photolithography technique utilizing a hydrophobic photoresist and SU-8 polymer material was introduced to fabricate microfluidic channels. The hydrophilic channels were precisely defined within hydrophobic walls, creating a pathway for liquid penetration. Capillary flow enabled the transportation of liquid through the hydrophilic channel and the paper matrix. Within the reaction site on the paper, specific reagents for glucose and protein assays were lithographically patterned in a distinctive three-branched tree design. This groundbreaking development significantly propelled research in this field. The simplicity and cost-effectiveness of this approach, coupled with its suitability for portable applications, have made it an attractive alternative to existing technologies [31,41,42].

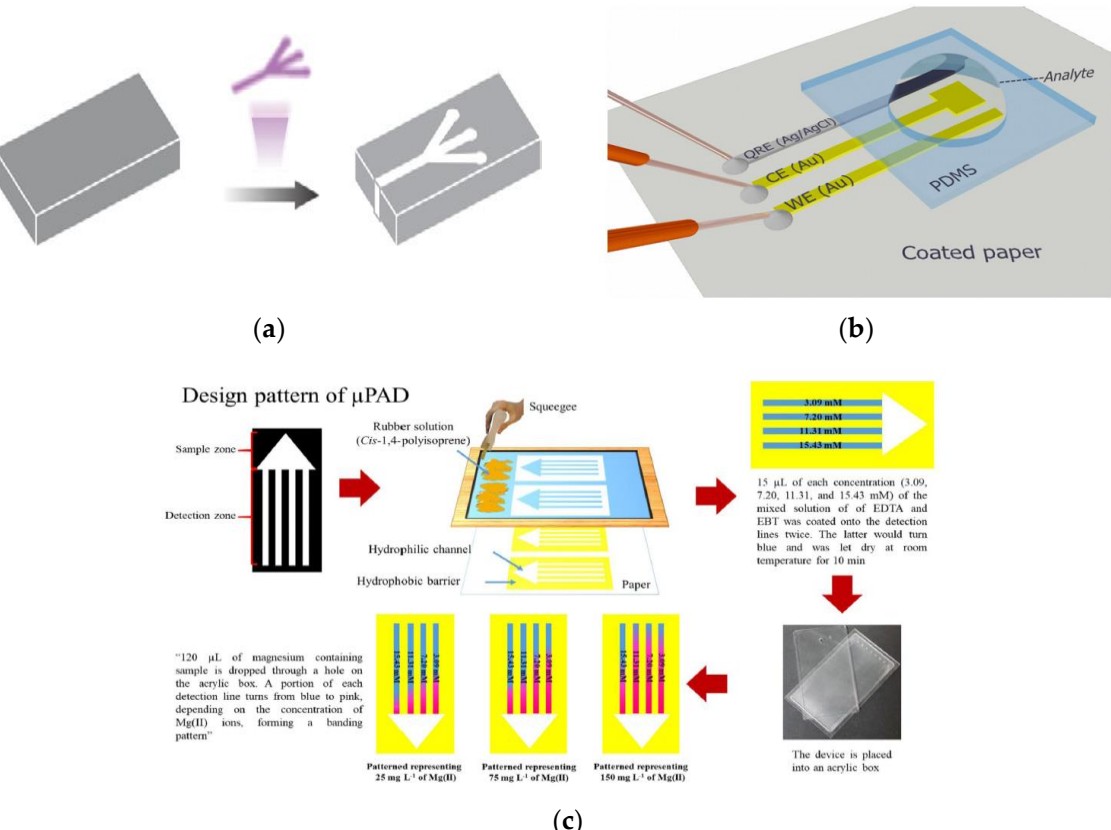

**Figure 1.** Schematics of fabrication: (**a**) photolithography, (**b**) inkjet printing, (**c**) screen printing [43–45].

Inkjet printing paper is patterned using a solvent-based printer and soaked in a polystyrene solution to make it hydrophobic. Inkjet printing toluene in a pattern removes the polystyrene from the paper. Pressing hydrophobic ink or wax through the screen transfers the desired pattern to the paper. This method can use more types of ink, but it requires a new screen for each pattern, making it unsuitable for rapid device prototyping.

Screen printing and photolithography patterns the design to be printed on a screen. There are other fabrication methods, including paper cutting, 3D printing, wax printing, chemical vapor deposition, and drop casting (shown in Figure 2) [46–48]. Paper cutting is the most traditional method, where paper is cut into a network of channels and zones using a knife cutter or laser cutter. Scissors or a knife can be used to cut paper to make simple devices. Tape or glass slides can be used to protect the cutting equipment and paper. Cutting is an easy process that does not require chemicals. However, devices without channels are more challenging to control and require a solid support for mechanical stability [46].

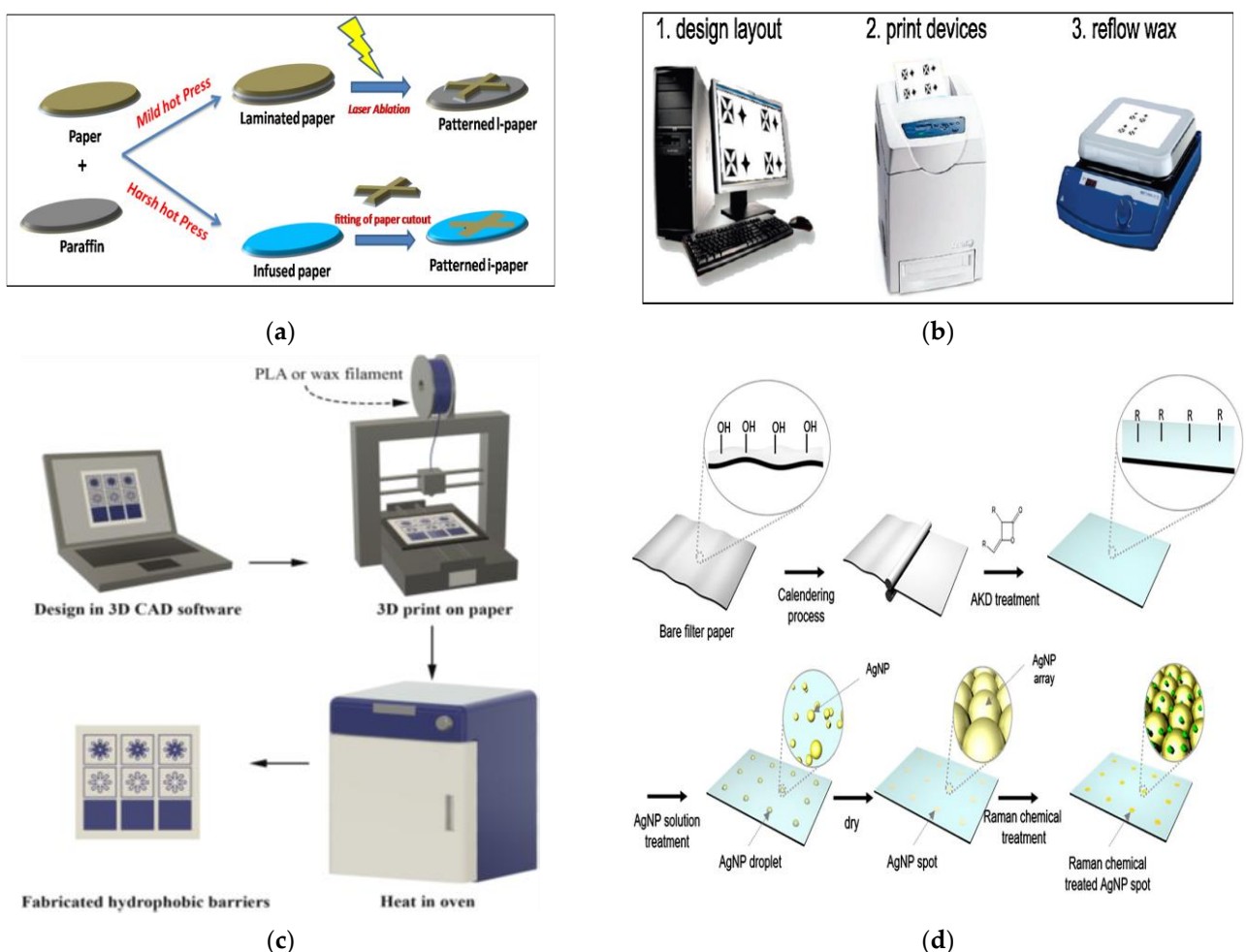

**Figure 2.** Schematics of fabrication (**a**) Paper cutting (**b**) Wax printing (**c**) 3D printing (**d**) Drop cast [46,47,49].

Wax printing has become a popular approach compared to the traditional paper-cutting method because of its simplicity, convenience, and speed. This method uses solid-ink printers that print molten wax on the surface of the paper. The heated wax spreads both laterally and vertically, which must be considered when designing the patterns. While the resolution of wax printing is adequate for the majority of applications, two methods of enhancing the technique's resolution have been introduced in recent years that produce sub-millimeter patterns on paper. 3D printing, also known as additive manufacturing, can aid in the production of 3D structures and complex geometry for rapid prototyping. This fabrication has been extensively used for paper-based microfluidics [46,47]. Next, a small-sized paper-based sensor is fabricated using the drop casting method by dropping the functional matrix solution onto the paper substrate and then evaporating the solvent [50].

## 5. Paper-Based Detection Methods

### 5.1. Optical Detection

Optical detection is the most common, least valuable, and fundamental method. Variability in illumination causes variations in color saturation; therefore, hue must be considered. This issue can be resolved by adjusting the white balance, removing the background color, or comparing the image to a calibration curve with known color and intensity standards. This is also applicable when multiple imaging devices are used, such as when comparing the results captured by a camera, a portable spectrometer, and a scanner. Extensive research has revealed that performing quantification on hydrophobic paper yields significantly more accurate outcomes compared to hydrophilic materials. Hydrophobic paper minimizes the spreading of reagent droplets, leading to smaller and more densely concentrated detection regions. This enhanced concentration improves the sensitivity of the assay and facilitates easier quantification. Conversely, experiments conducted on hydrophilic paper often encounter premature termination due to surface drying. Furthermore, the substrate's excessive absorption of sensing molecules can impede their accessibility. Various optical detection methods, such as colorimetry, fluorescence, and chemiluminescence, have been explored to achieve precise measurements on hydrophobic paper surfaces [28,51].

Optical detection is mostly used for biological detection [52]. A paper-based fluorescence and colorimetric glucose sensor (Figure 3) was developed by Yen-Linh Thi Ngo et al. [53] using nitrogen-doped carbon dots and hybrid metal oxide structures. In the presence of $H_2O_2$, the material exhibits inherent peroxidase-like activity, acting as a catalyst to oxidize 3,30,5,50-tetramethylbenzidine (TMB) to produce blue-emitting oxidized TMB (oxTMB), eliminating the requirement for glucose oxidase (GOx).

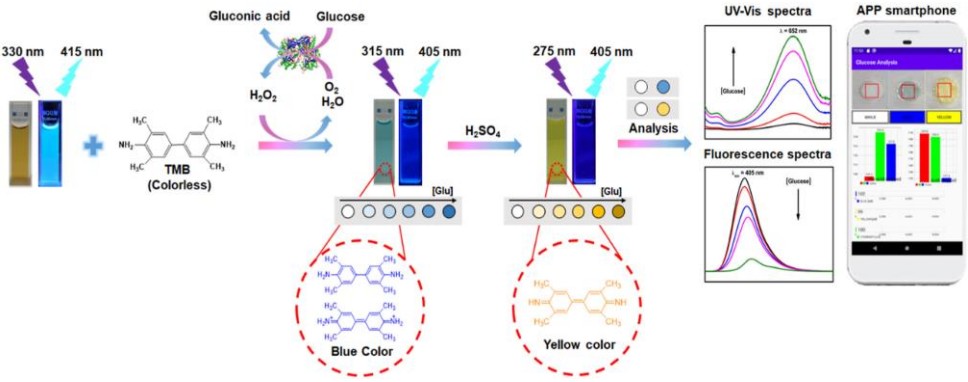

**Figure 3.** Schematic illustration of a paper-based sensor with optical detection [53].

### 5.2. Electrochemical Detection

Electrochemical sensors offer numerous advantages over optical sensors, as they are not affected by light, dust, or insoluble substances. Researchers have observed that applying a thin layer of liquid to the electrode surface can minimize the impact of fluid convection caused by random motion, vibrations, and heating. Furthermore, a static test can be converted into a flow measurement by connecting the paper channel's end to a cellulose blotting paper pad, facilitating continuous wicking between the electrodes. This setup is well-suited for convection, hydrodynamic measurements, electrode plating, and electrode cleaving. The design of paper platforms can also incorporate the back of the electrodes for three-phase electrolyte/electrode/gas interfaces. The large roughness and porosity of the deposited materials can increase the surface area, thereby enhancing the sensor's sensitivity. Various electrochemical detection techniques, including potentiometry, amperometry, voltammetry, and conductometry, can be employed in these systems [28,51].

Electrochemical detection is mainly used for biological detection [52]. Liu et al. [54] (Figure 4) introduced a versatile paper-based sensor platform modified with signal molecule-

labeled Deoxyribonucleic Acid (DNA) to detect biomarkers such as miRNA, ALP, and CEA. Specifically, they employed a microRNA-recognized probe for miR-21, a DNA aptamer probe for carcinoembryonic antigen, and a phosphorylated hairpin probe for alkaline phosphatase. These probes enabled the synthesis-dependent DNAzyme formation upon target recognition, leading to substrate DNA cleavage. The sensing system utilized target-triggered deposition and amplification of the DNAzyme-catalyzed signal, enabling highly selective zero-background detection.

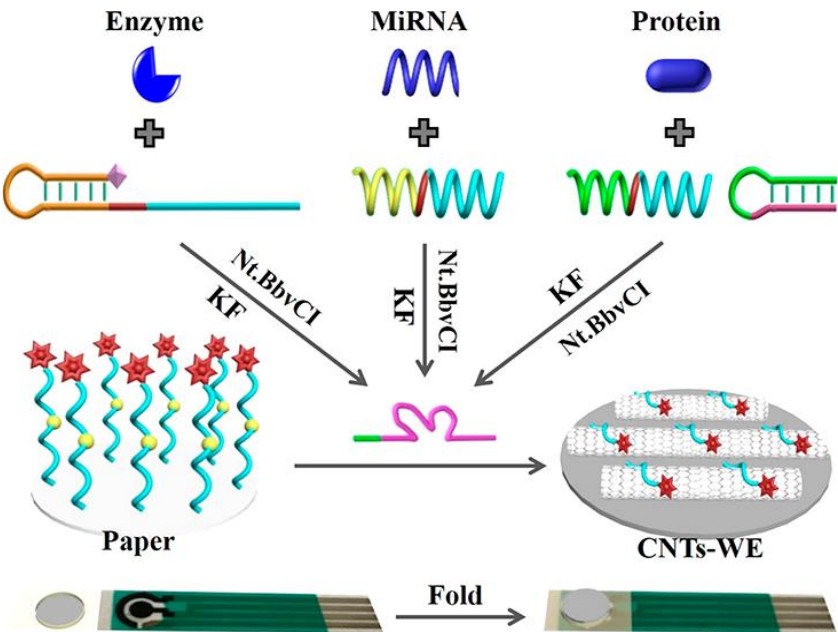

**Figure 4.** Schematic illustration of a paper-based universal electrochemical sensor system [54].

*5.3. Chemiresistive Detection*

Chemiresistive substances have been used to detect toxic chemicals and explosives [17]. There are two distinct types of chemiresistive sensors: metal oxide sensors and conductive polymer sensors. These sensors display different resistance values when exposed to various odors. Both types have the capability to form sensor arrays that consist of multiple sensors with different levels of sensitivity and selectivity. By integrating several microsensors with a resistive readout interface circuit on a compact substrate, it is possible to create a miniature sensor array.

Metal oxide sensors, in particular, offer the advantage of achieving a sensitivity level of approximately ten parts per million (ppm). Still, they also have disadvantages, such as a very high working temperature and the fact that the sensor can only detect elements at 300 °C. Although metal oxide sensors must operate at high temperatures, conductive polymer sensors can operate at room temperature (RT). An electronic interface sensor is simple, making it ideal for a portable instrument. The sensor's sensitivity can be as high as 15 ppm. The sensitivity to humidity is the main disadvantage, so it is essential to exclude background humidity and control sensor baseline drift when employing the sensors [55]. Chemiresistive detection can detect for gas, ion, and biologicals. When analytes interact chemically with the surface of a material, it causes a dynamic change in its electrical resistance (Figure 5). A marginal difference between the electrical resistance in the presence of an analyte and its absence indicates a stronger surface interaction [56].

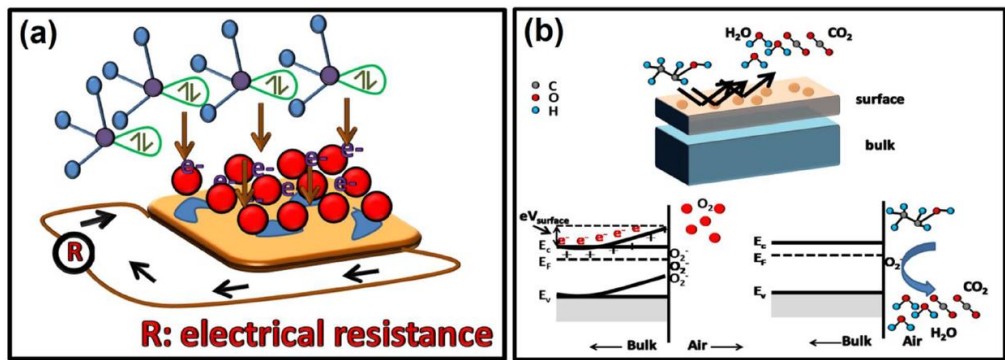

**Figure 5.** Schematic illustration of chemiresistive detection. (**a**) A visual depiction of an electrical circuit utilized for examining surface interactions. (**b**) The band bending mechanism elucidating the origin of sample resistance resulting from surface interactions [56].

## 6. Paper-Based Sensors to Detect Gas

There are numerous research reports on paper-based chemiresistive sensors to detect gas from various materials, such as metal oxide and/or inorganic materials, carbon-based materials, conductive polymers, and composite materials. Hirotaka Koga et al. [57] presented a disposable paper-based molecular sensor device that contains a ZnO nanowire (NWs) sensor to detect $NO_2$ gas. This paper substrate is made with wood-derived biodegradable cellulose nanofiber and uses a low-cost graphite electrode. This paper-based product was produced using a two-step papermaking procedure. Additionally, the NW structure helps to make the sensor's network strong and well-connected. This structure gives the sensor easy access to the target molecule and good electrical contact with the electrode. In addition, Hao Kan et al. [58] used lead sulfide (PbS) NWs sprayed into paper substrates at RT to detect $NO_2$ gas. The graphite electrode was drawn with a pencil to simplify the sensor's design and construction. PbS-NWs on a paper sensor have a microstructure porous network that can present efficient gas adsorption.

Some researchers focused on other gases such as $H_2$, $NH_3$, and $H_2S$. Abhishek Kumar et al. [59] used PdMoY alloy nanosheets (NS) to detect hydrogen ($H_2$) in the air, and the Pd alloy 2D solution-phase NS were prepared and then drop cast onto paper ($\approx 1 \times 1$ cm) on which silver contacts were drawn and dried. The same substance content was deposited on an interdigitated electrode (IDE). The rough and porous surface of the paper gives a higher response due to the fact that more gaps are formed between Pd NS.

Additionally, two researchers have reported on carbon-based chemiresistive materials for the detection of gas. They focused on $NH_3$ gas sensing with single-walled carbon nanotubes (SWCNT) and carbon nanotubes (CNT). Loukkose et al. [60] presented a straightforward approach for fabricating paper-based sensors to detect $NH_3$ vapor at room temperature (RT). The method involved depositing chemiresistors, which were purified and solution-processed single-walled carbon nanotubes (SWCNT), onto paper substrates using vacuum filtration. This material has a bundling size structure, and the sensing performance can be enhanced by decreasing the bundling size.

On cellulose paper, an SWCNT-based $NH_3$ gas sensor was developed by Jin-Woo Han et.al. [61] two types of device, a CNT-on-paper and a CNT–cellulose composite, were manufactured and compared. Due to the larger reaction surface, the CNT-on-paper exhibited rapid recovery/response and greater sensitivity than the CNT–cellulose composite. The cellulose fibers can also be used in chemical reactions due to the fact that the CNTs are intertwined not only with adjacent CNTs but also with the cellulose fibers. Multiple hydroxyl groups are present on the surface of the cellulose polymer. Some hydroxyl groups on the surface form hydrogen bonds with adjacent celluloses or CNTs. The paper-based sensor exhibited superior uniformity and reproducibility in comparison to the glass-based control sensor. The current method can be applied to smart papers with inexpensive disposable applications.

Additionally, several results on conductive polymers to fabricate paper-based sensors are listed below. The spraying technique was used for the first time to develop a flexible sensor based on flower-like polyaniline-coated filter paper (PCFP), enabling the non-contact and rapid detection of nitroaromatic explosives by Weiyu Zhang et al. [62]. The high sensitivity and rapid response to 2, 4, 6-trinitrotoluene (TNT), picric acid (PA), and 2, 4-dinitrotoluene (DNT) at room temperature were evident due to the hierarchical flower-like structures, which have high permeability of filter paper. This structure can effectively prevent the agglomeration of PANI fibers and produce large amounts of gas.

Xiao Ye et al. [63] presented ketjen black (KB) ink and molecularly imprinted sol–gel (MISG) inks for the fabrication of a fully inkjet-printed chemiresistive sensor array to detect volatile organic acids (VOAs). The MISG/KB sensor was fabricated on photographic paper with a three-layer structure. Hexanoic acid (HA), heptanoic acid, and octanoic acid were used as templates to manufacture the MISGs and as targets to assess the detection and discrimination capabilities of the sensor array. The three resulting MISG/KB sensors exhibited a high degree of sensitivity and selectivity towards VOA vapors. In addition, sonochemical polymerization of pyrrole with an oxidizing agent and tosylate co-dopant (TS) were used with cellulosic paper strips decorated with aminophenyl-modified multiwalled carbon nanotubes (MWCNT) to create inexpensive, conductive Pap@CNT-$NH_2$@PPy composites to detect $NH_3$ gas sensors by Ouezna Hamouma et al. [64]. The electrochemical properties of the paper electrodes were characterized using cyclic voltammetry and electrochemical impedance spectroscopy. This research demonstrates the combination of aryl diazonium-modified CNTs and in situ sonochemical polymerization on a paper surface as a functional material for gas sensors.

There are several research reports based on chemiresistance from a composite material (combination of two materials), such as robust boron nitride nanotube/carbon nanotube (BNCNT). Guh-Hwan Lim et al. [65] presented a vacuum filtration-fabricated thermally stable and hybrid paper for fully reversible self-enhanced chemiresistive sensing to detect $NO_2$ gas. Boron nitride nanotubes (BNNT) were crucial for durable reliability (33 days) at an operating temperature of 200 °C. To further comprehend the thermal behavior of the BNCNT network structure, the finite-element method was employed. The long-term thermal stability of BNNTs plays a significant role in improving the $NO_2$ sensing ability of the CNT paper.

Lianghui Huang et al. [66] used inkjet printing to produce poly (m-aminobenzene sulfonic acid) (PABS)-functionalized SWCNT for a paper-based sensor to detect $NH_3$. To fabricate the $NH_3$ gas sensor, dispersed SWCNT-PABS was printed onto Ag electrodes that had previously been printed. The rheological properties of SWCNT-PABS dispersion and sensor surface structures were analyzed. The paper-based sensor has a good sensing response, rapid recovery, and response time to varying concentrations of $NH_3$ at ppm levels. Song-Jeng Huang et al. [67] detailed the fabrication of inorganic nanotube (INT)—tungsten disulfide ($WS_2$), graphene—PEDOT: PSS sheet, and $WS_2$ nanotube-modified paper-based conductive chemiresistors for butanol gas sensing. The production of $WS_2$ nanotubes with a bundle structure required a two-step process involving oxide reduction and sulfurization at 900 degrees Celsius. The continuous coating produced the graphene–PEDOT: PSS (poly(3,4-ethylene dioxythiophene):poly(styrene sulfonate) hybrid conductive paper sheet. Chemiresistors were produced by depositing tungsten disulfide ($WS_2$) nanotubes onto conductive paper using a drop casting technique. The results indicated that the graphene–PEDOT: PSS/$WS_2$ NTs chemiresistor constructed from paper can detect butanol gas with high sensitivity, rapid recovery, and excellent reproducibility. Table 2 summarizes this section.

**Table 2.** Paper-based sensors to detect gas.

| Sensing Material | Type of Sensing Material | Type of Paper | Contact/Detect/Analyte | Morphology and Chemical Bonding | Result | Reference |
|---|---|---|---|---|---|---|
| ZnO | Metal Oxide | Cellulose Paper | $NO_2$ | Morphology:<br>• The ZnO NWs.<br>• Cellulose paper with a nanofiber structure.<br>Chemical Bonding: - | The paper-based sensor detected $NO_2$ at 98 ppm with a sensitivity of 9 (Equation (1)), and the detection was about 3.9 ppm. | [57] |
| PbS | Inorganic Material | Cellulose Paper | $NO_2$ | Morphology:<br>• PbS NWs.<br>Chemical Bonding: - | The sensitivity of the PbS NWs sensor to detect $NO_2$ was 17.5 (Equation (2)) with a response time of about 3 s and a recovery time of about 148 s at a concentration of 50 ppm. | [58] |
| PdMoY | Inorganic Material | Cellulose Paper | $H_2$ | Morphology:<br>• Pd alloy nanosheet.<br>• The paper has a rough and porous structure.<br>Chemical Bonding: - | The sensitivity of the paper-based PdMoY NS sensor was about 18.7% (Equation (4)). | [59] |
| SWCNT | Carbon Based Material | Filter Paper | $NH_3$ | Morphology:<br>• The SWCNT has a bundle structure.<br>Chemical Bonding: - | At 62.5 ppm NH3, the paper-based sensor exhibited fast response (30 s) and recovery (30 s) characteristics, and it was able to detect concentrations as low as 80 ppb. | [60] |
| CNT | Carbon Based Material | Cellulose Paper | $NH_3$ | Morphology: -<br>Chemical Bonding: -<br>• There were hydrogen bonds between cellulose molecule and CNT. | The minimum detection limit of the CNT-on-paper sensor was found to be 5 ppm. | [61] |
| PCFP | Conductive Polymer | Filter Paper | TNT, 2, 4-DNT, PA | Morphology:<br>• Polyaniline has a flower structure.<br>Chemical Bonding: - | • The PCFP sensor exhibited high sensitivity to TNT, PA, and DNT with approximate values of 237.1%, 100.6%, and 80.1% (Equation (5)), respectively.<br>• TNT, PA, and DNT had average response and recovery times that did not exceed 8.1 and 1.9 s, respectively.<br>• The limit of detection for TNT and PA was found to be very low, at 0.094 and 0.029 ppb (0.000094 and 0.000029), respectively. | [62] |
| KB/MISG | Conductive Polymer | Glossy Paper | VOAs | Morphology:<br>MISG has high porosity with mesoporous structure.<br>Chemical Bonding: - | The LOD was about 0.018 ppm. | [63] |
| Pap@CNT-NH$_2$@PPy (minophenyl-modified multiwalled carbon nanotubes) | Composite | Cellulose Paper | $NH_3$ | Morphology:<br>The surface of the cellulosic fibers was covered in aggregated nanostructures.<br>Chemical Bonding: - | • The sensitivity was about 525% (Equation (4)), with a limit of detection of 0.04 ppb (0.00004 ppm).<br>• The response and recovery times were about 138 s and 465 s, respectively. | [64] |
| BNCNT | Composite | Filter Paper | $NO_2$ | Morphology:<br>BNCNT papers have porous and fibrous structure<br>Chemical Bonding: - | The sensitivity of this paper-based sensor was 16.5% with LOD about 3.41 ppb (0.00341 ppm). | [65] |
| SWCNT-PABS | Composite | Glossy Paper | $NH_3$ | Morphology: -<br>Chemical Bonding: - | The sensitivity of this paper-based sensor was 201% (Equation (4)). | [66] |
| Graphene–PEDOT:PSS/WS$_2$NT | Composite | Filter Paper | Butanol | Morphology: The nanotube has a bundle structure<br>Chemical Bonding: - | The detection limit was 44.92 ppm, with a response time of 205 s and a recovery time of 20 s | [67] |

The table shows that not all researchers mentioned both physical and chemical properties. While previous papers have extensively covered the physical properties of paper-based sensors, there are a limited number of publications that specifically discuss their chemical properties. Many researchers have found that the structure of the sensing material has a greater impact on the sensor's performance than the paper's structure.

The sensitivity of each gas sensor is determined by a different formula (equation) than the one mentioned in Section 2. As a result, a direct comparison between the sensitivities of

these gas sensors is not possible. The smallest limit of detection for a paper-based sensor is found in Polyaniline-Coated Filter Paper (PCFP) for detecting TNT and PA. In addition, not all researchers have mentioned the sensitivity and/or limit of detection in their studies.

The NW structures can be found in Table 1. Previous research on ZnO and PbS, particularly, showed a high surface-to-volume ratio, which enhanced gas adsorption. Figure 6 shows the NW structure as observed with Scanning Electron Microscopy (SEM). The previous research presented in Table 1 revealed the presence of NW structures, particularly in ZnO and PbS, which exhibit a high surface-to-volume ratio, hence enhancing gas adsorption. Figure 7 displays various structures besides NWs, including nanosheets, bundles, and others observed by SEM. Nanosheets were detected in PdMoY as well. This nanosheet structure can improve sensor response due to the rough and porous surface of the paper, causing more gap formation between Pd nanosheets. Additionally, a paper-based sensor that utilizes carbon-based material has a bundle structure on SWCNT.

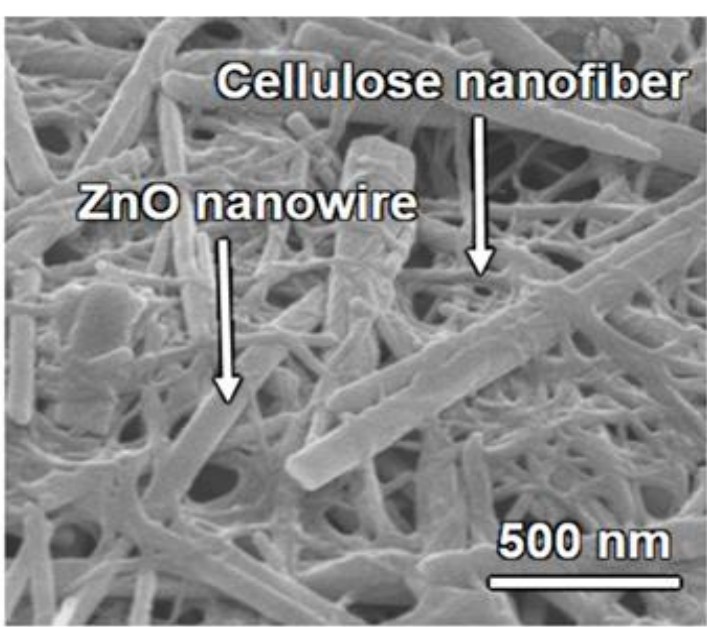

**Figure 6.** NW structure of ZnO by SEM [57].

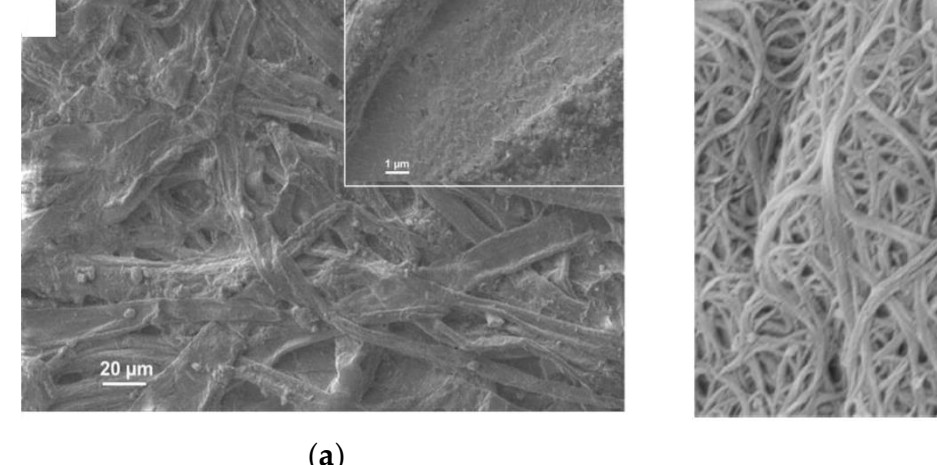

(**a**) (**b**)

**Figure 7.** *Cont.*

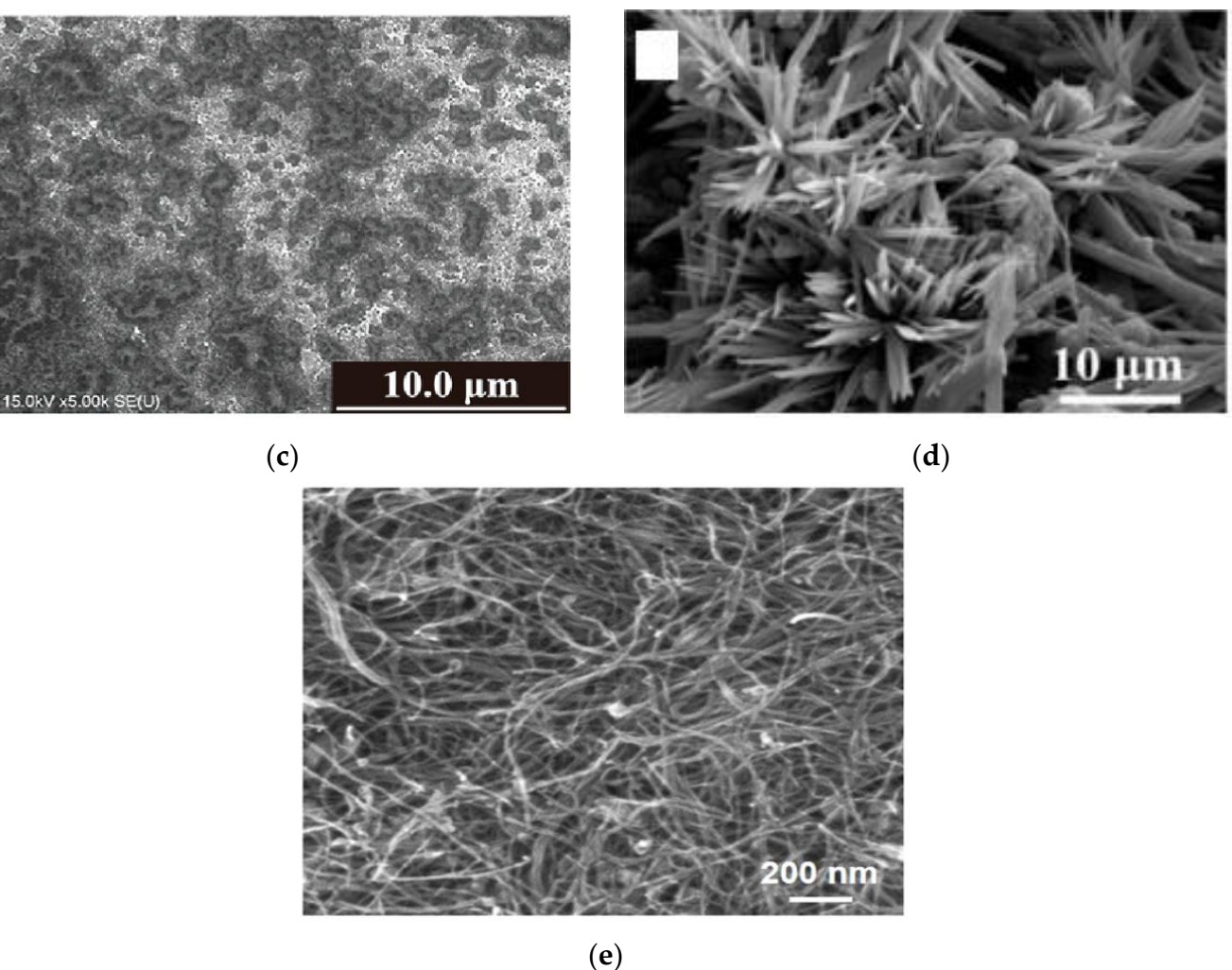

**Figure 7.** Various Structures: (**a**) Nanosheet Pd Alloy, (**b**) Bundle SWCNT, (**c**) Mesoporous MISG, (**d**) Flower Structure PCFP, (**e**) Fibrous Structures BNCNT papers [59,60,62,63,65].

Nanostructures have been extensively employed in sensing materials for paper-based sensors to detect gases, primarily because of their well-established capability to enhance performance [68]. Various sensing materials have specific reasons for employing nanostructures, according to metal oxide and inorganic material, the utilization of small crystallite sizes contributes to their improved performance. Additionally, the nanostructures have a high surface-to-volume ratio, evincing better sensing than bulk-size materials [69,70]. The sensing mechanisms of metal oxide gas sensors are based on changes in electrical conductivity, which are attributed to charge transfer between surface complexes, involving molecules such as $O^-$, $H^{2-}$, $OH^-$, and other interacting molecules [71].

In CNT, apart from good conductivity, the nanostructures on carbon can provide efficient exposure of surface groups for the bonding between analyte molecules and transduction material, leading to high detection of pollutants [69,72,73]. CNT-based gas sensors operate on the sensing mechanisms rooted in the p-type semiconductor properties. The transfer of electrons between carbon nanotubes (CNTs) and the oxidizing or reducing gas molecules adsorbed on their surface modifies the electrical conductance. Specifically, in the case of p-type CNTs, an increase in the number of adsorbed oxidizing gas molecules led to a decrease in electric resistance [74]. On the other hand, in conductive polymers, the nanostructure of the polymer has a high performance with a rapid response, and this is attributed to their electrical, catalytic, and thermal properties [75].

Chemical bonding can be found in the research of Jin Woo Han et al. [61]. Certain surface hydroxyl groups established hydrogen bonds with adjacent cellulose or carbon nanotubes (CNTs). However, there existed unconnected surface hydroxyl (OH) groups that could function as chemical reaction sites. Consequently, a chemical reaction occurred between the hydroxyl group and ammonia (as depicted in Equation (8)). It is worth noting that these reactions did not affect the resistance of cellulose since its fiber backbone inherently exhibited insulating properties. Nonetheless, the localized presence of cellulose fibers interspersed with carbon nanotubes (CNTs) enabled intertube charge transport, facilitating conduction despite the natural insulating characteristics of cellulose fibers.

$$OH + NH_3 \rightarrow NH_2 + H_2O \tag{8}$$

This review mainly focuses on the physical and/or chemical properties that influence the sensor. However, it is crucial to note that the mechanical properties of the sensors are closely linked to their durability. The ability of a sensor to withstand mechanical stress and maintain its functionality over time can be assessed through bending tests. By evaluating the sensors' resistance to bending and other mechanical stresses, their durability can be quantified. Thus, the mechanical properties of the sensors play a significant role in their overall performance and long-term reliability [76,77].

Several researchers who have reported on sensing materials often solely attribute sensor performance to factors such as sensitivity and/or limit of detection, without considering the impact of the sensors' mechanical properties. Several researchers, including Hao Kan et al. [58], Loukkose et al. [60], Weiyu Zhang et al. [62], Xiao Ye et al. [63], and Guh-Hwan Lim et al. [65], have reported the sensor's mechanical properties for detecting gas.

## 7. Paper-Based Sensors to Detect Ions

There are some research reports on the use of paper-based chemiresistive sensors to detect ions from various materials, e.g., inorganic materials, carbon-based materials, and composite materials. For inorganic materials, Rinky Shal et.al. [17] described a novel ultra-low-cost pyrite $FeS_2$-based smart sensor on a flexible paper substrate to detect Methyl jasmonate (MeJa) ions. This significant improvement in the analytical performance of the $FeS_2$ sensor can be attributed to the high conductivity of the pyrite $FeS_2$, the large surface area resulting from its microsphere-like morphology consisting of vertically grown NSs, and the presence of defects in the $FeS_2$ structure. Furthermore, wireless monitoring of MeJa was accomplished through the hydrothermal growth of pyrite $FeS_2$ on cellulose paper and its integration with a microcontroller, from which the collected data were transmitted to a smartphone via Bluetooth, thereby facilitating remote sensing. The invention of such a low-cost nanomaterial-based disposable sensor represents a significant advancement in the production of affordable lab-on-chip devices for analytical applications.

An (Ag-PAD)-based chemiresistor composed of silver ink for detecting $NO_2$ ions has been developed by Yu-Ci Liu et al. [78]. As a result of $NO_2^-$ initiating a diazo reaction and then reacting with ink and silver ink, it functions as an effective resistance-based transducer. Pulsed light sintering is used to synthesize the silver ink onto the PADs from nanoparticles (a combination of silver NWs and nanoparticles). Compared to other nanoparticle and paper-based sensors, this sensor for nitrite has several advantages, including an excellent linear range, a lower LOD, better stability, higher selectivity, low-volume sampling, and the fact that it is disposable. This paper-based sensor has been implemented to successfully determine $NO_2^-$ concentration in various places, involving taps, rivers, and lakes.

There have been some reports from carbon-based materials, a simple paper-based aptasensor for ultrasensitive $Pb^{2+}$ ion detection within 10 min has, been developed by Zahra Khoshbin et al. [79].Utilizing the Forster resonance energy transfer (FRET) process and the superfluorescence property quenched by graphene oxide (GO) sheets, the aptasensor has been successfully designed. When GO is added to the FAM-labeled aptamer, it stimulates noncovalent bonding via stacking and hydrogen bonding interactions between the aptamer's nucleotide bases and the aromatic structure of GO. As a result, the fluo-

rophore comes into proximity with the GO surface, resulting in fluorescence quenching of the aptamer via the energy transfer process. Subsequently, the $Pb^{2+}$ injection on the test zone causes the aptamer conformation to change from a random coil to a G-quadruplex structure, which is highly stable due to its compact design.

A novel enrichment-based paper test for distinguishing heavy metal ions was developed by Liang Feng et al. [80] compared to conventional paper-based microfluidic devices, and the sensitivity of the enrichment-based technique was greatly enhanced. Using our enrichment-based pyridylazo ($C_{15}H_{11}N_3O$) compounds array paper, we obtained the discrimination ability of eight different heavy-metal ions at the same concentration as low as 50 μM by combining eight pyridylazo compounds with array-based pattern recognition. Using a standard chemometric technique, the ions of heavy metals were easily identified. Obviously, this method can also be applied to other analytes.

Yi Kuang Yen et al. [81] introduced a paper-based nanohybrid chemiresistive sensor that can detect free chlorine ions ($Cl^-$) using a mobile phone with composite materials. The sensor was manufactured using a straightforward and standardized coating process. The addition of nanohybrid graphene with (PEDOT: PSS) to a paper-based sensing device resulted in a more stable and spontaneous response. The advantages of combining two materials for this paper-based method are portability, low cost, and ability to measure water quality. Next, the study conducted by Mohammad Rostampour et al. [82] introduced a combination of poly(3-octylthiophene) (POT) and single-walled carbon nanotubes (SWC-NTs) as sensing materials in a paper-based substrate for ion-selective electrodes (ISEs). The aim was to detect potassium ($K^+$) and sodium ions ($Na^+$). Filter paper was utilized in this research, and the resulting paper-based ISEs for potassium and sodium ions demonstrated outstanding sensor performance and remarkable reproducibility. Table 3 summarizes this section.

**Table 3.** Paper-based sensors to detect ions.

| Sensing Material | Type of Sensing Material | Type of Paper | Contact/Detect/Analyte | Morphology and Chemical Bonding | Result | Reference |
|---|---|---|---|---|---|---|
| Pyrite $FeS_2$ | Inorganic Material | Cellulose Paper | MeJa | Morphology:<br>• The pyrite $FeS_2$ has a microfiber structure.<br>Chemical Bonding: - | The sensor had a detection limit of 0.68 mM and showed good sensitivity of $12.24 \pm 14\%$ mM$^{-1}$ (Equation (6)) in the 1–2.5 mM range of MeJa. | [17] |
| Ag-PAD | Inorganic Material | Cellulose Paper | $NO_2$ | Morphology:<br>• Ag has nanoparticles structure.<br>Chemical Bonding: - | The LOD was $8.5 \times 10^{-11}$ M. | [78] |
| GO | Carbon-Based Material | Chromatography paper | $Pb^{2+}$ | Morphology:<br>-<br>Chemical Bonding: Hydrogen Bonding | The LOD was 0.5 pM. | [79] |
| $C_{15}H_{11}N_3O$ | Carbon-Based Material | Array Paper | $Hg^{2+}$, $Cd^{2+}$, $Pb^{2+}$, $Ni^{2+}$, $Cu^{2+}$, $Zn^{2+}$, and $Co^{2+}$ | Morphology: -<br>Chemical Bonding: - | The LOD was 50 μm. | [81] |
| Graphene and PEDOT: PSS | Composite | Filter Paper | $Cl^-$ | Morphology:<br>• Paper has porous structure.<br>Chemical Bonding: - | The paper-based sensor was able to detect chlorine in a linear range from 0.1 to 500 ppm, with a detection limit of 0.18 ppm. | [81] |
| POT and SWCNT | Composite | Filter Paper | $K^+$ and $Na^+$ | Morphology: -<br>Chemical Bonding: - | The LOD were $7.3 \pm 0.4 \times 10^{-7}$ ($K^+$) and $1.1 \pm 0.1 \times 10^{-6}$ M ($Na^+$) | [82] |

Based on this table, it was found that not all researchers mentioned both the physical and chemical properties of the paper-based sensor. Previous studies have indicated that the structure of the sensing material has a greater impact on the sensor's performance than the paper's structure. This is consistent with the development of paper-based gas sensors, and nanostructures commonly employed in sensing materials for paper-based sensors due to their demonstrated enhanced performance, as well as the previous explanation given in Section 6 [68].

The LOD of paper-based sensors for detecting ions may be expressed in different units, making direct comparisons difficult. Only one researcher mentioned the sensitivity of the sensor. Chemical bonding was observed by Zahra Khoshbin et al. [79]. The addition of graphene oxide (GO) to the FAM-tagged aptamer resulted in the formation of noncovalent bonds facilitated by interactions such as π-π stacking and hydrogen bonding between the nucleotide bases of the aptamer and the aromatic structure of GO. This interaction brought the fluorophore into close proximity to the surface of GO, leading to the quenching of aptamer fluorescence through Förster resonance energy transfer (FRET). Subsequently, the introduction of $Pb^{2+}$ in the test zone induced a conformational change in the aptamer, causing it to transition from a random coil structure to a highly stable G-quadruplex structure with a compact arrangement. In addition, none of the researchers mentioned in this section reported or mentioned the mechanical properties of the sensors. However, a discussion on the mechanical properties of sensors can be found in Section 6.

## 8. Paper-Based Sensors for Biological Detection

There are numerous research papers on paper-based chemiresistive sensors to detect biological samples from various materials, comprising inorganic materials, carbon-based materials, conductive polymers, and composite materials. Biological data such as those taken from urine and breath are also reported from inorganic materials. Mei-Lin Ho et al. [83] showed a paper-based analytical device (LE-PAD) that can find leukocyte esterase (LE) as a quantitative point-of-care test for urinary tract infections (UTI). The LE-PAD has a silver-conducting film covered with 3-(N-tosyl-L-alaninyloxy)-5-phenylpyrrole (PE) and 1-diazo-2-naphthol-4-sulfonic acid (DAS). The urine and LE react with the PE and DAS, and the products react with the silver coating, which changes the resistivity.

T. Leelasree et al. [84] described the first flexible MOF-based sensor for breath-sensing applications. By combining the high-porous HKUST-1 MOF structure with a conducting $MoS_2$ material, an electronic sensor on adjustable paper support was made that can be used to study cases of sleep apnea. A plausible mechanism has been suggested, and a prototype based on a smartphone has been made to show how the hybrid device could be used in real life. This study demonstrates the excellent potential for using MOFs in healthcare, emphasizing breath sensing and sleep apnea diagnosis.

Besides inorganic materials, there are carbon-based materials. Sushmitha Veeralingam and Sushmee Badhulika [85] implemented a new approach of wax deposition followed by vacuum filtration to design hydrophobic and hydrophilic channels for label-free, highly selective and sensitive cholesterol detection, and made a paper substrate-based biosensor coated with multiwalled carbon nanotubes (MWCNTs). The studies on morphological characterization showed that MWCNTs are evenly spread out on the paper substrate, with an average diameter of 15–20 nm. The outstanding response of the fabricated biosensor can be attributed to the modulation of electrical properties ($ChO_x$) because of the electrostatic gating effect and direct electron transfer between MWCNTs and cholesterol due to cholesterol oxidase bioconjugation.

Marta Pozuelo et al. [18] described a paper-based chemiresistor consisting of a network of SWCNTs and anti-human immunoglobulin G (antiHIgG). SWCNTs are exceptional transducers owing to their high sensitivity in terms of resistance changes caused by immunoreaction for detecting human immunoglobulin G (HIgG). As a consequence, the resistance-based biosensor detects concentrations at scales as small as the picomolar. The resultant paper-based biosensor is sensitive, selective, and employs a low-cost substrate and straightforward manufacturing steps.

Several research reports on paper-based chemiresistive polymers for the detection biological molecules. Jacopo Emilio Giaretta et al. [86] presented a paper-based 3D-printable sensor made from PEDOT:PSS. Horseradish peroxidase is an enzyme capable of interacting with $H_2O_2$ through oxidation. This technology is impedimetric, which greatly simplifies the fabrication process. The resulting ink is inkjet-printed onto filter paper, where the highly

porous microstructure of the cellulosic paper facilitates immobilization of both polymer and enzyme.

Fabrication, optimization, and analytic characterization of a paper-based chemiresistive biosensor for label-free immunosensing are described by Yu Shen et al. [87], who detected human serum albumin (HSA) with water-based ink synthesized from pyrene carboxylic acid (PCA) and single-walled carbon nanotubes (SWNTs) via a non-covalent-stacking interaction. The PCA/SWNTs ink concentration could reach ~4 mg mL$^{-1}$ and was stable for one month at RT. In addition, the cellulose fibers containing hydroxyls aided in connecting the SWNTs to an electrical network. This paper-based chemiresistive biosensor is designed for the rapid, sensitive, and selective detection of HSA, and its fabrication is straightforward.

Vincesco et al. [88] developed a paper-based sensor using composite materials for the detection of biological substances. In their study, they utilized a screen-printed device with a working electrode that was drop cast with a nanocomposite consisting of carbon black and gold nanoparticles. In electrochemistry, square-wave voltammetry is used. This sensor was augmented with Nafion to detect iron ions. Under optimized conditions, iron ions were detected with a LOD of 0.05 mg/L and a linearity of 10 mg/L in standard solution. Table 4 summarizes this section.

**Table 4.** Paper-based Sensors to Detect Biological.

| Sensing Material | Type of Sensing Material | Type of Paper | Contact/Detect/Analyte | Morphology and Chemical Bonding | Result | Reference |
|---|---|---|---|---|---|---|
| PE/DAS/Ag | Inorganic Materials | Cellulose Paper | Urinary Leukocyte Esterase | Morphology: Nanoparticles and NWs Chemical Bonding: - | The LOD was 1.91 ($\times 5.1$ U mg$^{-1}$ mL$^{-1}$, S/N = 3). | [83] |
| MoS$_2$ | Inorganic Materials | Cellulose Paper | Human Breath | Morphology: Porous Chemical Bonding: - | The sensor's response time was 0.38 s. | [84] |
| MWCNTs | Carbon based material | Filter Paper | Cholesterol | Morphology: Cellulose Microfiber Chemical Bonding: - | The LOD was 3.2 nM. | [85] |
| SWCNTs and anti-human immunoglobulin G (antiHIgG). SWCNTs | Carbon based material | Filter Paper | Human Immunoglobulin G (HIgG) | Morphology: - Chemical Bonding: - | The sensitivity for the range 0–6.3 pM was $-1.737 \pm 0.85$ nA/pmols L$^{-1}$ (Equation (6)) | [18] |
| Horseradish Peroxidase (HRP)—PEDOT:PSS | Conductive Polymer | Filter Paper | Hydrogen peroxide (H$_2$O$_2$) | Morphology: - Chemical Bonding: Carbons Double bonding | The LOD was 61.3 $\times 10^{-9}$ M. | [86] |
| PCA and SWNTs | Composite | Filter Paper | Human Serum albumin (HSA) | Morphology: Porous Structure Chemical Bonding: - | The LOD was 1 pM, and the sensitivity was 9.44% (Equation (6)). | [87], |
| CB-AuNPs nanocomposite | Composite | Cellulose Paper | Iron in Blood Serum | Morphology: - Chemical Bonding: - | The LOD was 0.05 mg/L. | [88] |

Distinguishing themselves from paper-based sensors used for gas and ion detection, paper-based biological sensors can be influenced by both the type of paper and the sensing material, with their physical and/or chemical properties impacting sensor performance. There are several various sensing material structures shown in Figure 8, as observed with SEM. Although not all researchers mention physical and/or chemical properties in their studies, there are some issues that can be discussed, like the type of paper and detection.

Mostly, different types of filter paper are used for detecting biological substances because of their good controllable properties, namely, particle retention and flow rate, but cellulose papers are better because they have better immobilization of biological substances like proteins [10]. Only one researcher in this section, Mei-Lin Ho et al. [83], carried out surface modification on a paper-based analytical device with a coating that enhanced the sensor's performance, and this result is related in Section 3. In addition, the research conducted by Mei-Lin Ho et al. [83] demonstrated that the utilization of nanostructures on PE/DAS/Ag film can enhance the performance of sensors [68]. Porous structures on the paper surface in the research conducted by Yu Shen et al. [87] have been shown

to enhance the performance of sensors, offering low limit of detection (LOD) and high sensitivity. Additionally, these structures provide increased capacity to accommodate more single-walled carbon nanotubes (SWNT) [89,90]. Chemical interactions were observed in the study conducted by Jacopo Emilio Giaretta et al. [86], where carbon bonds can influence the sensing mechanism. Carbon single bonds (C-C) and carbon double bonds (C=C) were observed.

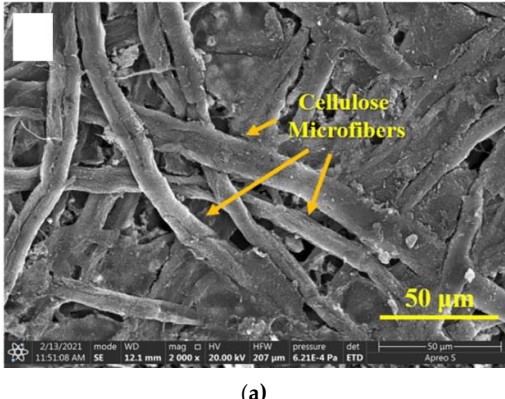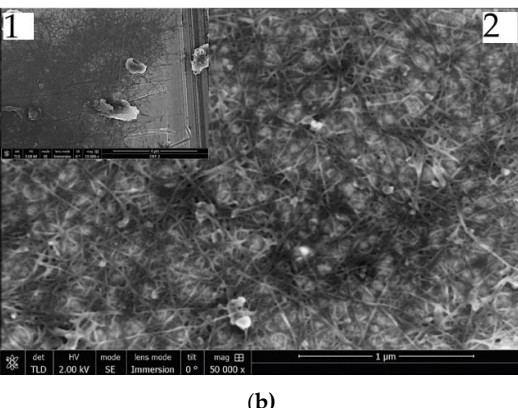

(**a**)      (**b**)

**Figure 8.** Various structures: (**a**) Cellulose microfibers on MWCNT, (**b**) (**1**) unmodified SWNTs/$H_2O$ at <0.1 mg mL$^{-1}$; (**2**) PCA/SWNTs in $H_2O$ at ~4 mg mL$^{-1}$ [85,87].

Table 5 displays the methods that can be used to detect biological material, and lists the majority of paper-based sensors that can detect biological material using electrochemical detection. This is because electrochemical detection has a higher sensitivity than optical detection, and electrochemical biosensors are widely acknowledged as a promising technique for the selective detection of analytes of interest [6,51,91]. In addition, none of the researchers in this section have reported or mentioned the mechanical properties of the sensors, just as in Section 7.

**Table 5.** Detection method for paper-based sensors to detect biological [18,83–88].

| Sensing Material | Detection Method |
|---|---|
| PE/DAS/Ag | Optical |
| $MoS_2$ | Chemiresistive |
| MWCNTs | Electrochemical |
| SWCNTs) and anti-human immunoglobulin G (antiHIgG). | Electrochemical |
| SWCNTs | |
| Horseradish Peroxidase (HRP)/(PEDOT:PSS) | Chemiresistive |
| PCA and SWNTs | Electrochemical |
| CB-AuNPs nanocomposite | Electrochemical |

## 9. Challenges and Future Perspectives

Paper-based sensing offers numerous advantages, but it also faces certain challenges. In the study conducted by Yu-Ci Liu et al. [78], an Ag-PAD-based chemiresistor was utilized for the detection of nitrite in various types of water, excluding seawater. However, in the future, there is potential for this chemiresistor to be employed in the detection of nitrite in seawater as well as other water sources. Several limitations and disadvantages associated with paper-based analytical devices (PADs) have been identified, including the time-consuming optimization process and lengthy fabrication procedures [51,92]. Moreover, there are limitations related to the efficiency of sample delivery within the device. Hydrophobic agents used for pattern devices may not provide sufficient hydrophobic barriers with low surface tension resistance. Additionally, the colorimetric method integrated into

the devices often results in a high limit of detection (LOD), rendering current paper-based microfluidic devices ineffective for the analysis of samples with low concentrations.

Sensitivity and LOD are critical parameters for evaluating the performance of a paper-based sensor. High sensitivity and low LOD are desirable, but the sensor must be durable for practical applications. Sensor durability is its ability to withstand environmental conditions, mechanical stress, and repeated use without affecting performance. A sensor with a low limit of detection (LOD) and high sensitivity may not be suitable for practical applications if it lacks durability or if its durability has not been adequately tested. To make paper-based sensors practical and reliable, sensitivity, LOD, and durability must be balanced [93,94].

During the COVID-19 pandemic that occurred between 2020 and 2022, paper-based sensors were utilized for the detection of SARS-CoV-2, the virus causing the disease. These sensors proved to be effective due to their high sensitivity, enabling accurate detection of the viruses [95,96]. Although paper-based sensors have high sensitivity, there are challenges such as regular standardization and approval for clinical use [97,98].

M. F. Zaki et al. [99] employed 3D printing to fabricating paper-based analytical devices, yielding superior results in terms of channel size compared to other fabrication methods. This research presents a promising alternative for the future fabrication of paper-based sensors. Additionally, Yu Chen et al. [100] utilized a machine learning algorithm, specifically the support vector machine method, to assess the accuracy of a paper-based optoelectronic sensor (referred to as the "paper nose") for detecting volatile gases in the air. This study demonstrates the potential of using machine learning for evaluating the accuracy of paper-based sensors. Further research opportunities in paper-based sensing include the development of cost-effective and portable sensors capable of detecting trace elements at low concentrations (down to FM). Despite the rapid expansion of paper-based sensor research, many unanswered questions remain. For instance, while most studies utilize a single type of filter paper, which has proven effective, there is scope for exploring alternative porous materials that are compatible with organic solvents [101].

After machine learning, to enhance detection capabilities, paper-based sensors can be seamlessly integrated with advanced technologies like smartphones [102–104]. The integration with smartphones offers several advantages, including easy accessibility, portability, and user-friendly operation [105,106]. By utilizing the smartphone interface, users can effortlessly interact with the sensor, gain insights from data analysis, and receive timely alerts or notifications based on the sensor's readings. This integration empowers users with a convenient and efficient means of observing and analyzing sensor data in real-time [107].

## 10. Conclusions

This review is about paper-based sensors made from metal oxides and/or inorganic materials, carbon-based materials, conductive polymers, and composite materials. Each of the materials has its own ability to sense elements, namely, gas, ions, and biologicals. Both the physical structure (such as nanoparticles, NWs, and porous materials) and chemical bonding can affect the performance of the sensor, but only the type of paper and sensing material can affect the performance for detecting biologicals.

Several challenges in paper-based sensors can lead to sensing disadvantages, including limitations in detection, low durability, and non-compliance with regulatory requirements. These challenges can hinder the performance and effectiveness of the sensors, impacting their practicality and reliability. In the future, paper-based sensors can be enhanced for better sensing by incorporating advanced techniques such as 3D printing. Additionally, the integration of paper-based sensors with machine learning algorithms and/or smartphones offers tremendous potential for real-time data analysis, remote monitoring, and intelligent decision-making.

**Author Contributions:** Conceptualization, S.-J.H. and Y.-K.Y.; validation, S.-J.H., Y.-K.Y., P.N.I. and Y.A.; formal analysis, S.-J.H. and P.N.I.; Resources, P.N.I.; writing—original draft preparation, Y.A.; writing—review and editing, Y.A., S.-J.H., Y.-K.Y. and P.N.I.; visualization, Y.A. and P.N.I.; supervision, S.-J.H. and Y.-K.Y. All authors have read and agreed to the published version of the manuscript.

**Funding:** This research received no external funding.

**Institutional Review Board Statement:** Not applicable.

**Informed Consent Statement:** Not applicable.

**Data Availability Statement:** Not applicable.

**Conflicts of Interest:** The authors declare no conflict of interest.

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
