# Peer review of "A Review of Paper-Based Sensors for Gas, Ion, and Biological Detection"

_coatings, doi:10.3390/coatings13081326_

Round 1
Reviewer 1 Report
This manuscript provides an essential survey of paper-based analytical platforms for chemo/bio sensing. Based on the interdisciplinary nature of the topic it should be of broad interest. The organization of the manuscript is satisfactory and deserves for publication in Coatings. However, the number of published review papers in this field is high. The authors should clearly demonstrate what the superiority of the current review compared with previous ones is? I have a few suggestions that the authors should consider before further processing:
1. Paper substrates should be compared with other substates especially glass materials in terms of cost, uniformity, and transparency, especially in optical-based systems. The authors should refer and cite related works for more information. For example, Nature Communications, 2022, 13(1): 5757; Biosensors and Bioelectronics, 2021, 174: 112825.
2. It is more interesting if the authors provide related description of how recognition units of the sensors immobilized on the paper substrates.
3. Which variables of the paper-substrates have influence on the recognition performance of the sensors? How these variables are controlled in synthesis condition? The related description should add to the manuscript.
4. Besides, in the different sections, some works are mentioned but it is unclear the reason behind such selection. From my point of view, some criteria should be fixed.
5. Most of sections lack critical analysis. Currently, this reads as "who did what". As such it has very limited value. The deep discussion should be added to the manuscript.
6. There are several grammatical errors and unclear sentences in the manuscript. Please polish the whole manuscript carefully.
7. Section “10. Conclusion” is very general and should be rewritten. Focus on current challenges and future perspectives. Avoid duplicate information provided in previous sections.
There are several grammatical errors and unclear sentences in the manuscript. Please polish the whole manuscript carefully.
Author Response
Review 1
This manuscript provides an essential survey of paper-based analytical platforms for chemo/bio sensing. Based on the interdisciplinary nature of the topic it should be of broad interest. The organization of the manuscript is satisfactory and deserves for publication in Coatings. However, the number of published review papers in this field is high. The authors should clearly demonstrate what the superiority of the current review compared with previous ones is? I have a few suggestions that the authors should consider before further processing:
Thank you for your comment : Because previous review papers only covered gas detection, this article will cover ion and biological detection as well. So, this paper has superiority.
- Paper substrates should be compared with other substates especially glass materials in terms of cost, uniformity, and transparency, especially in optical-based systems. The authors should refer and cite related works for more information. For example, Nature Communications, 2022, 13(1): 5757; Biosensors and Bioelectronics, 2021, 174: 112825.
Thank you for your comment : we have added the explanation in section 3
“ Paper has many important functions in human life, and not only for common functions as writing and drawing, but paper can be used as a substrate for sensors due to its inher-ently flexible nature [23]. Paper as a substrate has many excellent properties compared to other substrates, including good flexibility, low cost, eco-friendly, lightweight, and ex-pandability [24]. In comparison to other substrates like glass and silicon, which lack the flexibility of paper and polydimethylsiloxane (PDMS), paper offers a cost advantage over both. Additionally, while PDMS exhibits good flexibility, it is comparatively more expen-sive than paper [24,25]. Further comparisons between different substrates are presented in Table 1.
Table 1. Comparison of Paper as Substrate to Other Substrates [26]
|
Property |
Paper |
Glass |
Silicon |
(PDMS) |
|
Surface Profile |
Medium |
Very Low |
Very Low |
Very Low |
|
Flexibility |
Yes |
No |
No |
Yes |
|
Physical Structure |
Fibrous |
Solid |
Solid |
Solid, gas – permeable |
|
Surface to Volume Ratio |
High |
Low |
Low |
Low |
|
Fluid Flow |
Capillary action |
Forced |
Forced |
Forced |
|
Biodegradability |
Yes |
No |
No |
To Some Extent |
|
High-throughput Fabrication |
Yes |
Yes |
Yes |
No |
|
Cost |
Low |
Medium |
High |
Medium |
- It is more interesting if the authors provide related description of how recognition units of the sensors immobilizedon the paper substrates.
Thank you for your comment : We have added the explanation in Section 3 and only found it for cellulose paper.
“ Various techniques are employed to immobilize responsive materials in cellulose substrates, including dip coating, drop casting, and vacuum filtration. Achieving strong physical interactions or chemical bonds between the responsive material and the cellulose substrate is crucial for successful immobilization. Substrate modification might be neces-sary to enhance the retention of the sensing material. In the dip-coating method, cellulose substrates are immersed in a solution containing the desired recognition element for a specific duration, followed by solvent evaporation. Drop casting is another commonly utilized technique, where a small volume of solution (e.g., 1-10 μL) is deposited onto the cellulose substrate. However, it's important to note that the capillary-driven flow on the cellulose substrate during drop casting can lead to uneven distribution of the responsive material [33]. “
- Which variables of the paper-substrates have influence on the recognition performance of the sensors? How these variables are controlled in synthesis condition? The related description should add to the manuscript.
Thank you for your comment : We have already added this explanation.
“ Several variables can be controlled to ensure the production of high-quality paper substrates. Firstly, the selection of paper type, such as cellulose paper and filter paper, is crucial as each paper type possess unique properties including porosity and surface chemistry. Choosing the appropriate paper type based on the desired sensor performance allows for control over these variables. Secondly, surface modification such as coating can be applied to the paper substrate to enhance porosity, surface area, and surface chemistry, ultimately improving sensor’s performance. Lastly, the fabrication techniques employed in sensor production, such as dip coating, also impact the properties of the paper sub-strate. Controlling factors such as solution concentration, immersion time, and drying condition during fabrication, one can effectively regulate the porosity, surface area, and surface chemistry of the paper substrate. “
- Besides, in the different sections, some works are mentioned but it is unclear the reason behind such selection. From my point of view, some criteria should be fixed.
Thank you for your comment : We have already fixed it, as in Section 2.
- Most of sections lack critical analysis. Currently, this reads as "who did what". As such it has very limited value. The deep discussion should be added to the manuscript.
Thank you for your comment : We have added some explanation in several sections: Sections 2, 3, 6, 7, 8, and 9.
- There are several grammatical errors and unclear sentences in the manuscript. Please polish the whole manuscript carefully.
Thank you for your comment : We have corrected it.
- Section “10. Conclusion” is very general and should be rewritten. Focus on current challenges and future perspectives. Avoid duplicate information provided in previous sections.
Thank you for your comment : We have added more.
“ This review is about paper-based sensors made from metal oxides and/or inorganic materials, carbon-based materials, conductive polymers, and composite materials. Each of the materials has its own ability to sense the elements, namely, gas, ions, and biologicals. Both the physical structure (such as nanoparticles, NWs, and porous materials) and chemical bonding can affect the performance of the sensor, but only the type of paper and sensing material can affect the performance for detecting biologicals.
Several challenges in paper-based sensors can lead to sensing disadvantages, including limitations in detection, low durability, and non-compliance with regulatory requirements. These challenges can hinder the performance and effectiveness of the sensors, impacting their practicality and reliability. In the future, paper-based sensors can be enhanced for better sensing by incorporating advanced techniques such as 3D printing. Additionally, the integration of paper-based sensors with machine learning algorithms and/or smartphones offers tremendous potential for real-time data analysis, remote monitoring, and intelligent decision-making. “

Reviewer 2 Report
Before I start the actual review, I would like to say that the submitted manuscript may be a working copy where I could see many comments that must be the communication between the authors.
The paper reviews various paper-based sensors for detecting Gas, Ion and Biological signals. Firstly, the title of the paper sounds incomplete to me, and I suggest that the authors change the title. The introduction section is adequate though it could have been more focused.
In the background section, the performance factors are discussed only for gas sensing though the authors claim that the review encompasses gas, ion and biological signals.
Section 3- paper types could have been more specific an focussed.
The fabrication section is well explained.
Sections 5, 6 and 7 are adequately explained and are scientifically significant in terms of the review.
In section 8, the title is incomplete. detect biological - signal/specimen?
Section 9 is brief. It can be elaborated by looking into the current work carried out in this segment in 2021-22 as during the Covid pandemic research in this field has shown a significant rise.
Section 10 is the conclusion section. This section should be more to the point, and should bring out the crux of the paper. In the present form I find that the conclusion is the weakest section in the entire manuscript.
The quality of English is adequate. Some minor corrections are needed.
Author Response
Review 2
Before I start the actual review, I would like to say that the submitted manuscript may be a working copy where I could see many comments that must be the communication between the authors.
The paper reviews various paper-based sensors for detecting Gas, Ion and Biological signals. Firstly, the title of the paper sounds incomplete to me, and I suggest that the authors change the title. The introduction section is adequate though it could have been more focused.
In the background section, the performance factors are discussed only for gas sensing though the authors claim that the review encompasses gas, ion and biological signals.
Response : Thank you for your comment. We have corrected
“ There are several factors that must be considered in calculating the paper-based sen-sor's performance. The sensitivity of gas sensors can be determined by several equations. Firstly, the gas-sensing response (R) is defined as the ratio of the resistance values of the sensor in the detected gas to the resistance in the air.
The sensitivity of ion and biological sensors can be determined by measuring the change in resistance, which is expressed as the normalized values of resistances. In this measurement, ΔR represents the difference between the initial resistance (R-initial) and the final resistance (R-final). This calculation is represented by equation 6. “
Section 3- paper types could have been more specific an focussed.
Response : Thank you for your comment. We have added some explanation
“ Paper has many important functions in human life, and not only for common functions as writing and drawing, but paper can be used as a substrate for sensors due to its inher-ently flexible nature [23]. Paper as a substrate has many excellent properties compared to other substrates, including good flexibility, low cost, eco-friendly, lightweight, and ex-pandability [24]. In comparison to other substrates like glass and silicon, which lack the flexibility of paper and polydimethylsiloxane (PDMS), paper offers a cost advantage over both. Additionally, while PDMS exhibits good flexibility, it is comparatively more expen-sive than paper [24,25]. Further comparisons between different substrates are presented in Table 1.
There are several types of paper used as sensor substrates, including cellulose paper, filter paper, and others. Cellulose paper is derived from materials such as wood, straw, reeds, and waste paper. Furthermore, this type of paper is both cost-effective and envi-ronmentally friendly. Its high porosity, good roughness, biocompatibility, biodegradable and hydrophilic properties make it suitable for various sensor applications [27,28].Additionally, filter paper has excellent wicking ability, which makes it a popular selection for fabricating paper-based sensors. Whatman brand filter paper no. 1 is mostly used because of its standard-grade filter paper with medium retention and flow rate. Be-sides No. 1, there is No. 4: filter paper with larger pores, compared to the standard grade, is often selected to mitigate the potential restriction of capillary pores and hinders liquid penetration caused by the solvent-induced swelling of cellulose fibers [29]. Apart from cellulose and filter paper, possess chromatography paper. Chromatography is also popu-lar due to the smooth surface it exhibits, both sides are uniform, has a medium flow rate, and a 0.18 mm thickness, which enables compatibility with commercial printing ma-chines [30]. Both filter paper and chromatography paper are derived from cellulose [31,32].
Various techniques are employed to immobilize responsive materials in cellulose substrates, including dip coating, drop casting, and vacuum filtration. Achieving strong physical interactions or chemical bonds between the responsive material and the cellulose substrate is crucial for successful immobilization. Substrate modification might be neces-sary to enhance the retention of the sensing material. In the dip-coating method, cellulose substrates are immersed in a solution containing the desired recognition element for a specific duration, followed by solvent evaporation. Drop casting is another commonly uti-lized technique, where a small volume of solution (e.g., 1-10 μL) is deposited onto the cel-lulose substrate. However, it's important to note that the capillary-driven flow on the cel-lulose substrate during drop casting can lead to uneven distribution of the responsive material [33]. There are other types, such as glossy and array paper. Glossy paper is a substrate comprised of cellulose fiber and an inorganic filler [34,35]. In addition, an array of paper strips has been developed for the detection of several metal ions [36].
Several variables can be controlled to ensure the production of high-quality paper substrates. Firstly, the selection of paper type, such as cellulose paper and filter paper, is crucial as each paper type possess unique properties including porosity and surface chemistry. Choosing the appropriate paper type based on the desired sensor performance allows for control over these variables. Secondly, surface modification such as coating can be applied to the paper substrate to enhance porosity, surface area, and surface chemistry, ultimately improving sensor’s performance. Lastly, the fabrication techniques employed in sensor production, such as dip coating, also impact the properties of the paper sub-strate. Controlling factors such as solution concentration, immersion time, and drying condition during fabrication, one can effectively regulate the porosity, surface area, and surface chemistry of the paper substrate [37–39]. “
The fabrication section is well explained.
Sections 5, 6 and 7 are adequately explained and are scientifically significant in terms of the review.
In section 8, the title is incomplete. detect biological - signal/specimen?
Response : Thank you for your comment. We have changed the title
“ Paper-based Sensors for Biological Detection “
Section 9 is brief. It can be elaborated by looking into the current work carried out in this segment in 2021-22 as during the Covid pandemic research in this field has shown a significant rise.
Response : Thank you for your comment. We have added this topic in section 9
“ During the COVID-19 pandemic that occurred, between 2020 and 2022, paper-based sensors were utilized for the detection of SARS-CoV-2, the virus causing the disease. These sensors proved to be effective due to their high sensitivity, enabling accurate detection of the viruses [90,91]. Although paper-based sensors have high sensitivity, there are challenges such as regular standardisation and approval for clinic use [92,93].“
Section 10 is the conclusion section. This section should be more to the point, and should bring out the crux of the paper. In the present form I find that the conclusion is the weakest section in the entire manuscript.
Response : Thank you for your comment. We have added more words.
“ This review is about paper-based sensors made from metal oxides and/or inorganic materials, carbon-based materials, conductive polymers, and composite materials. Each of the materials has its own ability to sense the elements, namely, gas, ions, and biologicals. Both the physical structure (such as nanoparticles, NWs, and porous materials) and chemical bonding can affect the performance of the sensor, but only the type of paper and sensing material can affect the performance for detecting biologicals.
Several challenges in paper-based sensors can lead to sensing disadvantages, including limitations in detection, low durability, and non-compliance with regulatory requirements. These challenges can hinder the performance and effectiveness of the sensors, impacting their practicality and reliability. In the future, paper-based sensors can be enhanced for better sensing by incorporating advanced techniques such as 3D printing. Additionally, the integration of paper-based sensors with machine learning algorithms and/or smartphones offers tremendous potential for real-time data analysis, remote monitoring, and intelligent decision-making.”

Reviewer 3 Report
1 Authors imported erroneous pieces of text from other papers [11].
When discussing formula (1), the authors call to the value S sensitivity, although in fact sensitivity is the derivative of response with respect to concentration.
2 In the title of the article, the authors combine the concepts of “Paper-Based sensors” and “Chemiresistive Sensor”, however, the references contain information about either Paper-Based sensors or Chemiresistive Sensor, since Paper-Based sensors are quite rarely Chemiresistive Sensor.
3. 70 citations is not enough for a review article. In addition, most of the references do not refer to the concept of Paper-Based Chemiresistive Sensor.
Author Response
Review 3
1 Authors imported erroneous pieces of text from other papers [11].
When discussing formula (1), the authors call to the value S sensitivity, although in fact sensitivity is the derivative of response with respect to concentration.
Response : Thank you for your comment. We have corrected the source, and we have checked sensitivity definition.
“ Sensitivity indicates a change in the physical and/or chemicalproperties of the sensitive material in the presence of gas. It is determined as the ratio of sensor’s resistance in the atmosphereof the target gas to its resistance in the air if the target gasis anoxidizing one. “
2 In the title of the article, the authors combine the concepts of “Paper-Based sensors” and “Chemiresistive Sensor”, however, the references contain information about either Paper-Based sensors or Chemiresistive Sensor, since Paper-Based sensors are quite rarely Chemiresistive Sensor.
Response : Thank you for your comment. we will change the title.
“ A Review of Paper-Based Sensor for Gas, Ion, and Biological Detection “
- 70 citations is not enough for a review article. In addition, most of the references do not refer to the concept of Paper-Based Chemiresistive Sensor.
Response : Thank you for your comment. We have added more citations than 80.

Round 2
Reviewer 3 Report
Not all of my comments were taken into account in the revised version of the article.
The authors still confuse the concepts of sensitivity and sensor response (2).
Author Response
Not all of my comments were taken into account in the revised version of the article.
The authors still confuse the concepts of sensitivity and sensor response (2).
Response : Thank you for your comment. We have already made revision again in this section 2
"
- Background
Generally, effectiveness of paper-based sensors, depend on several factors such as sensitivity, response time, recovery time, and the limit of detection. Sensitivity pertains to the sensor's ability to detect the specific chemical species within the desired range of interest. Response time is usually stated as the amount of time needed to reach 90% of the final value, measured from the time the measured variable's step input changed. Recovery time is the time taken by the sensor to return to its initial value after the target gas concentration has decreased to zero. The limit of detection (LOD) is determined by a statistical model that utilizes both simulated and experimental data, and is calculated based on a calibration curve [13–15].
2.1 Performance factors
The paper-based sensor's performance requires consideration of several factors. Gas sensor sensitivity can be evaluated by examining the sensor's resistance-to-air ratio. Additionally, when dealing with oxidizing gases, sensitivity (S) can be calculated using Equation 1. "